# UV-B Radiation Disrupts Membrane Lipid Organization and Suppresses Protein Mobility of GmNARK in *Arabidopsis*

**DOI:** 10.3390/plants13111536

**Published:** 2024-06-01

**Authors:** Qiulin Liu, Tianyu Wang, Meiyu Ke, Chongzhen Qian, Jiejie Li, Xi Huang, Zhen Gao, Xu Chen, Tianli Tu

**Affiliations:** 1Fujian Provincial Key Laboratory of Haixia Applied Plant Systems Biology, College of Life Science, Fujian Agriculture and Forestry University, Fuzhou 350002, China; liuqiulinbio@163.com (Q.L.); w171416@163.com (T.W.); meiyukey@outlook.com (M.K.); gaozhen0695@fafu.edu.cn (Z.G.); 2Horticultural Plant Biology and Metabolomics Center, Haixia Institute of Science and Technology, Fujian Agriculture and Forestry University, Fuzhou 350002, China; 3State Key Laboratory of Cellular Stress Biology, School of Life Sciences, Xiamen University, Xiamen 361102, China; qiancz2019@xmu.edu.cn (C.Q.); xihuang@xmu.edu.cn (X.H.); 4Beijing Key Laboratory of Gene Resource and Molecular Development, College of Life Science, Beijing Normal University, Beijing 100875, China; jiejieli@bnu.edu.cn

**Keywords:** UV-B, GmNARK, endocytosis, lipid organization

## Abstract

While it is well known that plants interpret UV-B as an environmental cue and a potential stressor influencing their growth and development, the specific effects of UV-B-induced oxidative stress on the dynamics of membrane lipids and proteins remain underexplored. Here, we demonstrate that UV-B exposure notably increases the formation of ordered lipid domains on the plasma membrane (PM) and significantly alters the behavior of the *Glycine max* nodule autoregulation receptor kinase (GmNARK) protein in *Arabidopsis* leaves. The GmNARK protein was located on the PM and accumulated as small particles in the cytoplasm. We found that UV-B irradiation interrupted the lateral diffusion of GmNARK proteins on the PM. Furthermore, UV-B light decreases the efficiency of surface molecule internalization by clathrin-mediated endocytosis (CME). In brief, UV-B irradiation increased the proportion of the ordered lipid phase and disrupted clathrin-dependent endocytosis; thus, the endocytic trafficking and lateral mobility of GmNARK protein on the plasma membrane are crucial for nodule formation tuning. Our results revealed a novel role of low-intensity UV-B stress in altering the organization of the plasma membrane and the dynamics of membrane-associated proteins.

## 1. Introduction

Light is a multifaceted resource for plants, acting as both an energy source through photosynthesis and a crucial signaling agent that influences their growth and development. Solar radiation encompasses a spectrum of wavelengths, including ultraviolet (UV) radiation, which spans from 200 to 400 nanometers, and photosynthetically active radiation (PAR), ranging from approximately 400 to 700 nanometers. Of the total solar radiation that reaches the Earth’s surface, about 7% to 9% falls within the UV spectrum [1,2,3]. 

UV radiation is further categorized into three types: UV-C (100–280 nm), which is the most harmful form but is entirely absorbed by the atmosphere; UV-B (280–315 nm), which passes through the stratospheric ozone layer, with only a fraction reaching the earth’s surface; and UV-A (315–400 nm), which is not absorbed by the ozone layer and reaches the earth’s surface in full. Consequently, UV-B represents the shortest-wavelength component of sunlight that actually reaches the earth. UV-B radiation plays a dual role in plant life, acting as both an environmental stressor and a developmental signal [4,5]. It is instrumental in regulating photomorphogenesis, which includes processes such as the inhibition of hypocotyl elongation, the expansion of cotyledons, and the accumulation of flavonoids. However, exposure to high-intensity UV-B can be detrimental to plants, causing DNA damage, alterations in membrane composition, protein cross-linking, and the formation of reactive oxygen species (ROS), leading to oxidative stress [6]. 

Plant responses to UV-B radiation are highly variable and depend on the radiation’s intensity and variability. This variation in response is a testament to the evolutionary adaptability of plants, which have honed their ability to sense and adapt to different UV-B environments. Consequently, plants have developed a suite of protective structures and mechanisms. Among these protective measures, the induction of antioxidant systems is particularly important, as they are the first line of defense against the oxidative stress that arises from UV-B exposure [7,8]. Oxidative stress is recognized as a central component of UV-B stress, with significant implications for the induction of damage across various cellular components, including DNA, proteins, and lipids [9,10,11]. The intricate ballet of antioxidants is essential in a plant’s stress defense repertoire, adeptly adjusting gene expression, fine-tuning the composition of membrane lipids, and mediating the activities of proteins and thioredoxin systems [11,12]. While these mechanisms offer a starting point for understanding, the detailed influence of UV-B stress on the dynamic interplay of membrane lipids and proteins still warrants further exploration. It is clear, however, that UV-B irradiation can trigger observable changes in membrane components and inflict oxidative harm on the plasma membrane (PM) under controlled experimental settings [13,14]. The underlying processes by which UV-B remodels plasma membrane functions and modulates the dynamics of integral proteins are areas that are not yet fully charted and require continued research and attention.

Different intensities of UV-B light increase nodule numbers in soybean: as shown in studies, low UV-B (0.01–0.055 mWcm^−2^) led to a 115% increase in nodule numbers; medium-intensity UV-B (0.03–0.1 mWcm^−2^) increased nodule numbers by 150%; and high UV-B (0.05–0.18 mWcm^−2^) increased nodule numbers by 132% [15]. Apparently, UV-B signaling and stress pathways fine-tune legume nodule formation [15]. Although it has been confirmed that UV-B fine-tunes soybean nodulation mainly via GmUVR8-GmSTF3-mediated photo-signaling and flavonoid biosynthesis [15], further research is needed to understand how UV-B light modulates leguminous nodulation. The leucine-rich repeat receptor-like kinase GmNARK (glycine max nodule autoregulation receptor kinase) mainly functions in the leaves and is one of the most important regulators of the AON (Autoregulation of Nodulation) in soybean [16]. Moreover, salt-stress-induced reactive oxygen species production triggers plasma membrane internalization [17]. GmNARK is located on the PM, and its function may be modified via endocytosis-mediated vesicle trafficking induced by NaCl treatment [18]. UV-B stress results in oxidative-stress-related reactions; therefore, GmNARK is an ideal research object for exploring UV-B-induced protein dynamics changes. Furthermore, UV-B irradiation significantly increased nodule primordium density by 194% in the WT and by 142–144% in *Gmnark* mutants, indicating that *Gmnark* mutants remain partially responsive to UV-B light [15]. These results indicate that the NARK-AON signaling module also plays a significant role in UV-B-stimulated hyper-nodulation, but the corresponding molecular mechanism remains undiscovered, and this mechanism may explain why UV-B radiation triggers the GmNARK protein to enhance nodulation.

This study is designed to broaden our understanding of the effects of UV-B radiation on the functionality of GmNARK. To this end, the research objectives are delineated as follows: (i) investigate whether UV-B radiation alters the organization of lipid rafts in the plasma membrane and (ii) examine the impact of UV-B on the dynamics of GmNARK around the plasma membrane and its endocytic trafficking pathways.

## 2. Results

### 2.1. UV-B Irradiation Increases the Proportion of Ordered Lipid Phase 

Genetic studies, superresolution imaging, and transmission electron microscopy have demonstrated that *Arabidopsis* Remorin proteins are crucial for plasma membrane nanodomain assembly [19]. An examination of the subcellular distribution of AtRemorin1.2 showed that the AtRemorin1.2-GFP signal on the PM was significantly enhanced by UV-B treatment (Figure 1A,B). Compared to WL treatment, the Remorin1.2-GFP fluorescence signal significantly increased under supplemental UV-B light (40 μW/cm^2^) applied for 24 h or over the long term (4 d) (Figure 1B). These data consistently prove that UV-B irradiation promotes lipid organization.

To understand whether lipid nanodomain organization is influenced by UV-B irradiation, we examined the effect of UV-B on lipid order by staining WT *Arabidopsis* cotyledons with the fluorescent probe di-4-ANEPPDHQ, which is able to sense the dipole potential changes of membrane lipids, with peak emission wavelengths ranging from 630 nm (the liquid-disordered phase, non-nanodomain) to 570 nm (the liquid-ordered phase, nanodomain) [20,21]. Compared with the WL-grown seedlings, di-4-ANEPPDHQ displayed a significantly higher generalized polarization (GP) value [22] after different intensities of UV-B irradiation were applied, indicating that UV-B enhances the formation of the ordered lipid phase (Figure 1C,D). Methyl-β-cyclodextrin (mβcd) is an efficient raft-disrupting agent and operates by depleting sterol from the membrane [23]. To examine the possible relationship between UV-B light and lipid raft organization, we treated the plants with WL/UV-B and the raft-disrupting agent mβcd separately or in combination. Compared with contrast, di-4-ANEPPDHQ displayed a significantly blocked generalized polarization (GP) value (Figure 1E,F). These data consistently prove that UV-B light influences lipid raft organization and enhances ordered lipid formation.

### 2.2. UV-B Impairs GmNARK Protein Dynamics on the PM

GmNARK is the homolog of the *Arabidopsis* CLAVATA1 (AtCLV1) receptor kinase [16]. Both AtCLV1 and GmNARK proteins localize to the PM and traffic within the endomembrane system [18,24]. To visualize GmNARK protein dynamics, we constructed a vector that fused the GmNARK coding sequence with the green fluorescent protein (GFP) gene and stably transformed it into *Arabidopsis*. Most of the GmNARK-GFP proteins accumulated on the PM, and small GmNARK-GFP particles were visualized in the cytoplasm (Figure 2A). This result suggests that GmNARK proteins are most likely trafficked between membrane and endosomal vesicles, consistent with previous findings [18]. 

To understand whether the lateral diffusion of GmNARK-GFP on the PM is influenced by UV-B light, we examined the molecular dynamics of cell-surface-distributed GmNARK-GFP at the single-particle level (nanoscale) via VA-TIRFM. The surface-distributed GmNARK-GFP particles with high signal intensity consisted of two types of GmNARK-GFP populations: highly mobile and non-mobile (Figure 2B–D). Compared with WL-grown seedings, UV-B-irradiated seedings showed a greater proportion of non-mobile GmNARK-GFP particles. Therefore, the overall mean-squared displacement (MSD) and diffusion coefficient of GmNARK-GFP were significantly attenuated by UV-B treatment (Figure 2E,F). Interestingly, the bright and mobile GmNARK-GFP particles were actively trafficked from the cell surface to the cytoplasm along actin-filament-like structures. The application of an actin-depolymerizing drug, latrunculin B (LatB) [25], and an endocytosis assembly inhibitor, Tyrphostin23 (Tyr23) [26], both toned down the dynamics of the bright GmNARK-GFP particles (Figure 2B,C,F), supporting the notion that GmNARK-GFP particles are trafficked along the cytoskeleton and within the endomembrane system. Compared with the bright GmNARK-GFP foci, the GmNARK-GFP particles with weak intensity were much less mobile. Quantification of the MSD and diffusion coefficient showed that UV-B irradiation also reduced the lateral mobility of weak GmNARK-GFP foci (Figure 2D,F). Taken together, these cellular visualizations prove that UV-B impairs the lateral diffusion of surface GmNARK proteins, which might be the consequence of the disruption of lipid raft organization.

### 2.3. UV-B attenuates GmNARK Protein Internalization from the PM

Endocytic internalization and recycling of membrane-associated proteins maintain the dynamic molecular distribution on the PM. A large number of membrane-associated proteins are continuously cycled and recycled via clathrin-mediated endocytic internalization, thus enabling efficient acquisition of signals and materials from the cell surface [27]. The disruption of lipid raft organization by UV-B treatment prompted us to further examine the general CME process in WL- and UV-B-treated seedlings of transgenic *Arabidopsis* lines that expressed clathrin-light-chain 2 (CLC2)-RFP. The lifetime of CLC2 on the PM represents the resident period of CLC2 particles, which were quantitatively detected using variable-angle total internal reflection fluorescence microscopy (VA-TIRFM). UV-B irradiation significantly enhanced the CLC2 resident time on the PM (Figure 3A,B), indicating that UV-B light decreased the efficiency of surface molecule internalization via CME. Thus, UV-B irradiation might generally suppress the endocytic trafficking of membrane-associated proteins.

Brefeldin A (BFA) has been widely used as an inhibitor of intracellular protein trafficking that blocks exocytosis but allows endocytosis [28]. To examine the dynamics of GmNARK proteins in the endomembrane system, we co-treated GmNARK-GFP seedlings with WL/UV-B light and BFA. BFA bodies were clearly visualized under continuous WL conditions, while the internalized GmNARK-GFP proteins within BFA bodies were reduced by 53% and 35% upon UV-B treatment for 24 h and 48 h, respectively (Figure 3C,D). Furthermore, UV-B irradiation resulted in the accumulation of a proportion of GmNARK-GFP proteins along the membrane (Figure 3C), indicating that UV-B light decreases the efficiency of GmNARK protein endocytosis.

Owing to the synergistic effect of UV-B irradiation on the lateral diffusion and endocytic trafficking of GmNARK molecules, the less mobile GmNARK particles might result in the disruption of GmNARK protein activity. Therefore, we detected GmNARK-GFP protein levels using Western blot. UV-B irradiation significantly decreased GmNARK-GFP protein levels after 5 days of exposure in both the total protein and microsomal fraction extracts (Figure 3E,F). Therefore, UV-B irradiation impairs the mobility of GmNARK protein, eventually leading to the attenuation of GmNARK protein levels. 

## 3. Discussion

Plants have developed a variety of strategies for coping with UV-B radiation, which is instrumental in their adaptation to their ever-changing environment. Despite constituting a minor fraction of sunlight, UV-B radiation can induce substantial developmental shifts in plants [29]. Our study reveals that UV-B radiation modifies the structure of plasma membrane lipid rafts and influences the protein dynamics of GmNARK.

UV-B irradiation has been shown to alter membrane integrity and causes oxidative damage to the PM, potentially leading to a loss of barrier function or membrane constituents [13,14]. Membranes contain abundant lipid raft constituents that act as platforms or scaffolds for crosstalk among membrane-associated protein complexes [30,31]. The organization and compartmentalization of lipid raft nanodomain structures in the PM determine the constrained and diffusive behavior of membrane surface molecules [32,33]. Remorin proteins, which are highly concentrated in lipid nanodomains and lipid raft nanodomain structures in the PM, determine the constrained and diffusive movement of membrane-associated molecules [32,33,34,35]. This study shows that UV-B irradiation enhances the AtRemorin1.2 signal and significantly increases AtRemorin1.1 protein abundance in the PM (Figure 1 and Figure 3), suggesting a positive effect of UV-B on lipid organization. Additionally, upon subjection to UV-B irradiation, the di-4-ANEPPDHQ displayed a significantly higher generalized polarization (GP) value [22], and lipid rafts were segregated into a more tightly packed, liquid-ordered (Lo) phase as well as a less tightly packed liquid-disordered (Ld) phase [36,37], indicating that UV-B enhances the formation of the ordered lipid phase (Figure 1). In addition, the Mßcd treatment significantly restored the promotional effect of UV-B on lipid order (Figure 1). These data indicate that UV-B stress influences lipid raft organization and enhances ordered lipid formation.

Membrane lipid organization influences endocytic internalization and coordinates the lateral diffusion of surface molecules to maintain the dynamic interactions among membrane proteins [38]. GmNARK protein, located on the cell membrane, likely participates in vesicular trafficking via the endocytic pathway [18]. GmNARK proteins predominantly accumulated on the PM, and small GmNARK particles mainly accumulated in the cytoplasm (Figure 2). This result suggests that GmNARK proteins are most likely trafficked between membrane and endosomal vesicles, consistent with the previous findings [18]. In this study, we investigated the GmNARK protein dynamics on the PM in response to UV-B irradiation and found that UV-B impaired the lateral diffusion of GmNARK proteins, which might be a consequence of the disruption of lipid raft organization (Figure 2). Lipid raft organization is crucial for the functioning of biological membranes [23], but how UV-B affects these rafts requires further study. A UV-B-induced highly ordered lipid is likely to also impair CME by delaying the participation of the necessary endocytic constituents (Figure 3). Changes in the liquid-ordered phase caused by perturbing cholesterol levels are also known to modulate the invagination of clathrin-coated pits and CME [39]. The detailed mechanisms of REM assembly are worthy of further investigation. Endocytic internalization and recycling of membrane-associated proteins maintain the dynamic molecular distribution on the PM. The dynamic movement and accumulation of receptor proteins in different sites are helpful for providing a rapid response to extracellular signal molecules [28]. BFA treatment causes the accumulation of internalized membrane-associated proteins within BFA compartments [40]. UV-B irradiation caused GmNARK protein bodies to remain along the membrane, a process that typically requires internalization (Figure 3), indicating that UV-B irradiation decreases the efficiency of GmNARK protein endocytosis.

Furthermore, UV-B irradiation significantly decreased GmNARK protein levels (Figure 3), suggesting that UV-B irradiation impairs the mobility of GmNARK protein and leads to the attenuation of GmNARK protein levels. In soybean, GmNARK functioned as a receptor kinase and the most important regulator in the AON pathway, and the *Gmnark* mutant is less sensitive to UV-B’s effect on nodulation [15,16]. This research confirms that UV-B irradiation modifies GmNARK protein dynamics, impacting its endocytic trafficking and overall protein levels, thereby influencing the AON pathway. Recent studies have clarified the nature of several UVR8-mediated and UVR8-independent pathways that regulate UV-B stress tolerance [6]. Here, we reveal the broader implications of UV-B radiation with respect to membrane lipid organization, endocytosis, and protein dynamics on the PM.

## 4. Materials and Methods 

### 4.1. Plant Materials and Growth Conditions

The wild-type *Arabidopsis thaliana* ecotype *Col-0* was used in this study. Some of the mutant and transgenic lines used in this study were described previously: *uvr8-6* [41], AtRemorin1.2-GFP [19], and clathrin-light-chain 2 (CLC2)-RFP [38]. Seeds of *Arabidopsis thaliana* were sown on 0.8% agar containing 1/2 Murashige and Skoog media at 22 °C under 16 h light/8 h dark photoperiod. WL (measured using an HR-350 Light Meter; Hipoint, Kaohsiung, Taiwan) and WL supplemented with UV-B light (measured using a UV-297 UV-B Light Meter; HANDY, Jiangmen, China) treatments were devised. In the UV-B treatment, the seedlings were allowed to grow for 4 days before they were subjected to UV-B radiation.

### 4.2. Used Primers, Vectors, and Cloning Strategy

Total RNA was extracted from soybean (*Williams 82*) leaves by using a TRIzol kit according to the user’s manual (Invitrogen, Catalog no.12183555). Total RNA (1 μg) was treated with DNaseI and used for cDNA synthesis using a SuperscriptIII RT kit (Invitrogen, Catalog no. 18080093). We cloned the cDNA region of GmNARK (*Glyma.12G040000*) via PCR (forward primer: GGGGACAAGTTTGTACAAAAAAGCAGGCTTAATGAGAAGCTGTGTGTGCTAC; reverse primer: GGGGACCACTTTGTACAAGAAAGCTGGGTTGA GATTAATTAGGTTGTGAGTGTGAGTAG), and GFP was cloned into pGWB605 using Gateway clone technology; the resultant *35s:GmNARK-GFP* construct was introduced into *Arabidopsis thaliana Columbia-0*. Transgenic plants were generated via the floral dip method [42] using *Agrobacterium* strain *GV3101*.

### 4.3. Quantification of Remorin1.2-GFP Signal 

WL-grown 4-day-old pAtRemorin1.2:AtRemorin1.2-GFP *Arabidopsis* seedlings were placed under UV-B light (40 μW/cm^2^) for 24 h or for long-term treatment (4 d) or continuously grown under WL. Remorin1.2-GFP signal was captured using Zeiss LSM880 (Zeiss, Germany) in airyscan mode under uniform settings. The signal profile at the PM was tracked by plotting a line along the PM. Remorin1.2-GFP signal was automatically tracked using Image J (Plot Profile).

### 4.4. Lipid Staining

Lipid order visualization was performed in the leaf cells of 7-day-old *Arabidopsis* seedlings. For lipid staining, 5 μM final concentration of Di-4-ANEPPDHQ was added to the medium for 5 min staining before visualizing membrane lipid order. After being washed 3 times in 1/2 × MS medium, the stained root samples were mounted for observation using a Leica SP8 (Leica, Germany) confocal laser scanning microscope with an Alpha Plan Apochromat 63x, NA 1.4 oil objective [19]. For Methyl-β-cyclodextrin (mβcd) treatment, seedlings were incubated with mβcd (with a final concentration of 10 mM, liquid 1/2 MS as a solution) for 24 h treatment at room temperature [19]. Then, the treated seedlings were stained using di-4-ANEPPDHQ.

### 4.5. Quantification of Lipid Polarity

The Di-4-ANEPPDHQ fluorescence signals were excited at 488 nm and recorded simultaneously in the range of 500–580 nm for membrane order phase and 620–750 nm for membrane disorder phase. The GP (Generalized Polarization) value at each point was obtained using the averages of all valid pixels recorded in each region of interest (ROI) of the images. Di-4-ANEPPDHQ-stained signals were calculated using ImageJ as described in [22]. The ordered (500–580 nm) and disordered (620–750 nm) phase fluorescence images were denoted as ch00 and ch01, respectively. Macros were subsequently incorporated into the calculation of the GP value for each pixel that correlated positively with the membrane lipid order extent. The threshold value for the analysis was fixed at 15, and the color scale for the output GP images was set to ‘16 colors’; no immunofluorescence mask was selected. After the processing, ROIs (Regions of Interest) were manually selected from GP images for further mean GP value calculations. More than 60 ROIs from at least 10 images for each treatment were selected to generate the mean GP value data. Representative images display the merged images from the green (500–580 nm) and red (620–750 nm) emission channels of di-4-ANEPPDHQ (merge pictures). The radiometric color-coded GP images were generated in HSB (Hue–Saturation–Brightness) pictures, in which the GP value was assigned to hue (color), and the mean intensity was set to brightness [19].

### 4.6. Microscopic Observation and Data Quantification

Seedlings were mounted on 0.8% agar 1/2 MS chamber slides or liquid 1/2 MS glass slides containing the indicated concentration of chemicals and then immediately imaged. Images were taken using Zeiss LSM 880 (Zeiss, Germany) with Airyscan or Leica SP8 (Leica, Germany) confocal microscopes. The settings for excitation and detection were as follows: GFP: 488 nm, 505–550 nm; red fluorescent protein (RFP): 554/561 nm, 565–650 nm. All images in a single experiment were captured using the same setting. The quantification method is described below.

CLC2 lifetime: CLC2-RFP single particles were captured using VA-TIRFM included with the Zeiss Elyra PS.1 system (Zeiss, Germany). Time-lapse images spanning up to 120 s were acquired by using the Zeiss Alpha Plan Apochromat (Zeiss, Germany)100× (NA = 1.46) oil objective. CLC2-RFP lifetimes were then quantified by extracting the particle kymographs in ImageJ. Lifetimes of more than 30 particles from at least three seedlings were analyzed.

GmNARK diffusion velocity and efficiency: GmNARK-GFP particle time-lapse images with 50 or 200 ms exposure times and no intervals were captured using the same microscope used for CLC2 lifetime. The time-lapse images were first cropped into 15 μm × 15 μm (bright particles) or 5 μm × 5 μm (weak particles). Regions of Interest (ROIs) were processed using ImageJ; then, the cropped ROIs were imported to SpatTrack [43] for single-particle tracking and quantifying the *GmNARK-GFP* dynamic. The particle diameter was set to 0.6 μm for the following particle intensity analysis. The estimated particle diameter and the intensity threshold were set to 0.6 μm and 96%, respectively, for particle detection. The largest displacement between each frame was set to 0.9 μm, and the gap-closing event was not allowed to ensure the precise extraction of the particle trajectories. The final tracking data with a more-than 3 s time course were extracted to plot the trajectories and further used for MSD quantification in SpatTrack by using the following equation:$$MSD(t)=\frac{1}{M-n}\sum_{i=1}^{M-n}\left\{{\left(x\left(i+n\right)-x\left(i\right)\right)}^{2}+{\left(y\left(i+n\right)-y\left(i\right)\right)}^{2}\right\}$$
in which *x* and *y* represent the particle’s location, *M* indicates trajectory length with respect to the image frame, and n is the frame number corresponding to *t*. The Diffusion Coefficient was further analyzed by using the equation MSD(*t*) = 4 Dt, in which *D* represents Diffusion Coefficient, and MSD(*t*) was obtained by fitting the original MSD curve with the browning diffusion model. For each treatment, ROIs from at least ten time-lapse images taken from more than three seedlings were analyzed. 

### 4.7. BFA Treatment and Quantitative Analyses

BFA was dissolved in dimethyl sulfoxide to reach a concentration of 1 M. Seven-day-old *Arabidopsis* seedlings were incubated with BFA (200 µM, liquid 1/2 MS as a solution) for 30 min at room temperature [44]. GFP fluorescence was observed with a confocal laser-scanning microscope (Leica SP8 (Leica, Germany) confocal). Quantitative analyses of vesicles were performed using the corresponding confocal images. The internalization signal of GmNARK-GFP, termed the BFA body, was quantified by comparing the GFP signal intensity of the BFA body with the GFP signal on the PM. For each seedling, the vesicle number was calculated from the average of 10 independent leaf cells, with 30 seedlings per line measured for each treatment. ROIs from at least 10 cells were analyzed for each treatment. 

### 4.8. Western Blotting

Total protein content was extracted from *Arabidopsis* seedlings in protein extraction buffer (50 mM Tris–HCl (pH 7.5), 150 mM NaCl, 1 mM EDTA, 10% glycerol, 0.1% Tween 20, 1 mM phenylmethylsulfonyl fluoride, and complete protease inhibitor cocktail (Roche)) [45]; microsomal protein was extracted from *Arabidopsis* seedlings according to the protocol in [46]. Protein was extracted from 1 g *Arabidopsis* samples and tested using SDS-PAGE gel. Samples were immunoblotted with individual primary antibodies, namely, α-GFP antibody (Invitrogen, Catalog no. MA5-15256, 1:1000), anti-Remorin1.1 antibody (amino acid: ESEKSKAENRAQC) (1:1000), and anti-RPN6 [47] antibody (1:1000), and the secondary antibody ECL anti-Rabbit IgG (GE healthcare) (1:5000). Quantification of protein levels: protein signal strength of GmNARK-GFP, REM1.1, and RPN6 was calculated and given in arbitrary units using ImageJ (Plot Profile).

### 4.9. Statistical Analysis, Image Analysis, and Figure Preparation

Statistical data were analyzed using Graphpad Prism 7 (GraphPad Software, La Jolla, CA, USA). Statistical analyses were performed using Unpaired Two-tailed Student’s *t*-test assuming equal variances, where one, two, three, and four asterisks (*) correspond to significant differences with *p*-values of ≤0.05, ≤0.01, ≤0.001, and ≤0.0001, respectively, while “ns” represents non-significant difference. Camera and confocal images were prepared with ImageJ (http://imagej.nih.gov/ij/) or SpatTrack [43].

## 5. Conclusions

Plants perceive UV-B radiation as both an environmental cue and a potential stressor influencing growth and development. Legume plants display a unique response to UV-B by increasing the number of nodules. GmNARK also plays a significant role in UV-B-stimulated hyper-nodulation in soybean, but the corresponding molecular mechanism is not fully understood. We have shown that UV-B radiation increased the proportion of ordered lipid phase and disrupted clathrin-dependent endocytosis. This disruption, in turn, affects the endocytic trafficking and lateral mobility of the GmNARK protein on the plasma membrane in Arabidpsis leaves, suggesting a nuanced role of GmNARK in refining legume nodule formation under UV-B stimulation. Here, we provide a novel insight into how UV-B alters plasma membrane lipid raft organization and its potential implications for plant responses to UV-B radiation. Further studies are required to fully understand the impact of GmNARK proteins on nodulation in response to UV-B radiation.

## Figures and Tables

**Figure 1 plants-13-01536-f001:**
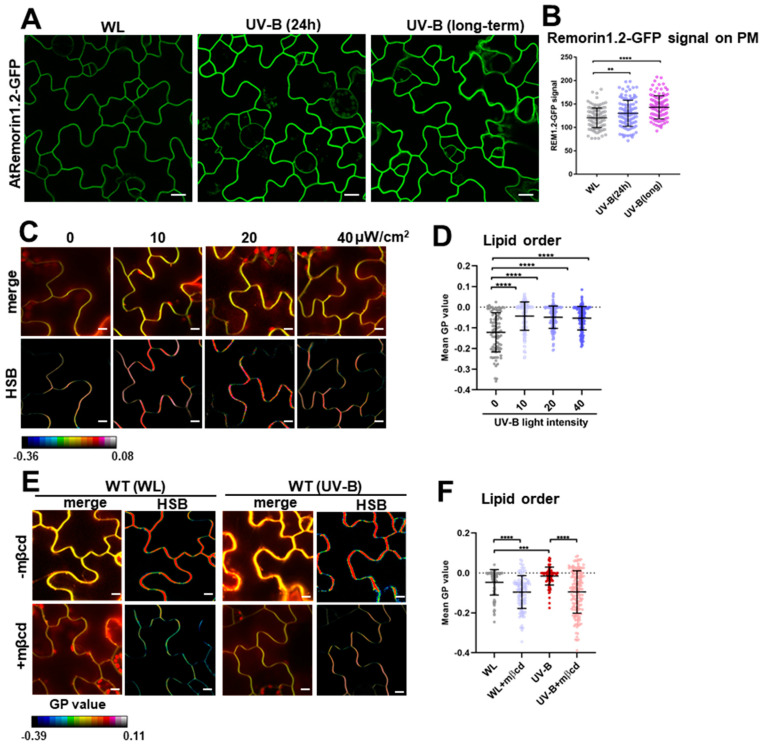
UV-B light increases the proportion of ordered lipid phase and disrupts clathrin-mediated endocytosis. (**A**,**B**) WL-grown *pAtRemorin1.2:AtRemorin1.2-GFP* (in *Arabidopsis*) seedlings were exposed to UV-B light (40 μW/cm^2^) for 24 h or in long-term (4 d) treatment or continuously grown under WL. The Remorin1.2-GFP fluorescence signal on the PM was calculated (from left to right: n = 118, 128, and 123 cells (**B**)). (**C**–**F**) PM lipid order was visualized in *Arabidopsis* leaf epidermal cells in a series of UV-B treatments (**C**,**D**). The seedlings were treated with WL or UV-B (40 μW/cm^2^, 24 h) in the absence or presence of 10 mM of mβcd (24 h) (**E**,**F**). Then, the treated seedlings were stained with di-4-ANEPPDHQ. Radiometric color-coded GP images were generated in HSB pictures (**C**,**E**), and mean GP value was calculated (from left to right: n = 88, 147, 165, and 156 images (**D**); n = 85, 110, 88, and 148 images (**F**)). Scale bars, 10 μm (**A**,**C**,**E**). Error bar = S.D. *p*-values were determined using two-tailed Student’s t-test, assuming equal variances (** *p* < 0.01; *** *p* < 0.001; **** *p* < 0.0001).

**Figure 2 plants-13-01536-f002:**
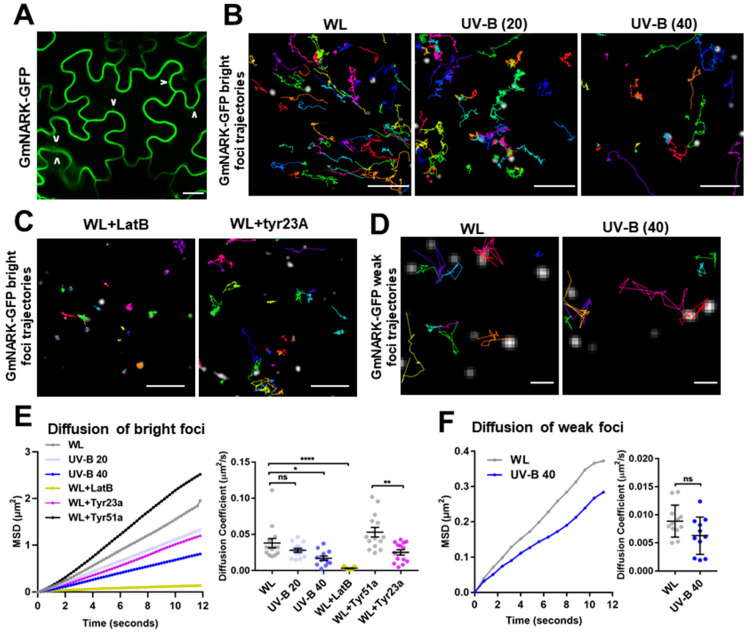
UV-B disrupts the internalization and lateral diffusion of GmNARK proteins. (**A**) GmNARK-GFP distribution of WL-grown *35s:GmNARK-GFP* (in *Arabidopsis*) was visualized. The internalized GmNARK-GFP proteins within the cytosol are highlighted by arrows. (**B**–**F**) GmNARK-GFP foci on the PM were recorded using VA-TIRFM in 5-day-old *35s:GmNARK-GFP* (in *Arabidopsis*) seedlings under WL, UV-B light (20 or 40 μW/cm^2^ for 48 h), WL supplemented with LatB (10 μM, 1 h), or WL supplemented with Tyr23A/51A (30 μM, 1 h). Cotyledon epidermal cells were observed. Trajectories of GmNARK-GFP bright foci (**B**,**C**,**E**) and weak foci (**D**,**F**) were individually tracked. Mean-squared displacement (MSD) and diffusion coefficient of GmNARK-GFP particle are plotted and quantified in (**E**) (left panel: n = 60 images; right panel: n = 14, 14, 12, 11, 14, and 14 images), (**F**) (left panel: n = 15 images; right panel: n = 12 and 11 images). Scale bars, 10 µm (**A**), 5 µm (**B**,**C**), and 1 µm (**D**). Error bar = S.D. *p*-values were determined using two-tailed Student’s t-test, assuming equal variances (* *p* < 0.05; ** *p* < 0.01; **** *p* < 0.0001; ns, not significant).

**Figure 3 plants-13-01536-f003:**
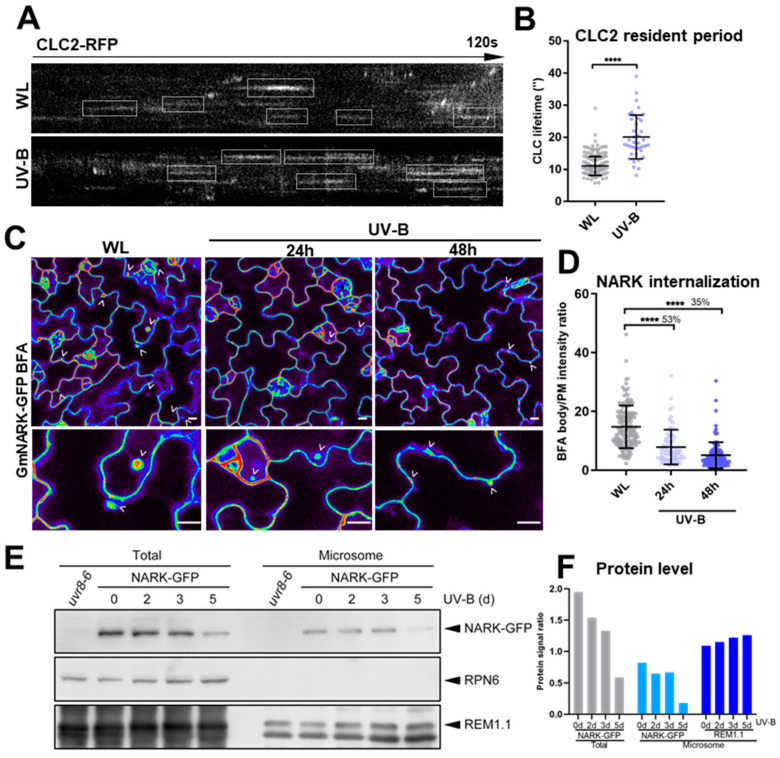
UV-B impairs mobility of GmNARK proteins and attenuation of GmNARK protein levels. (**A**,**B**) CLC2-RFP lifetime on PM was quantitatively measured in WT *Arabidopsis* leaf epidermal cells. 5-day-old WT seedlings were treated with WL or UV-B (40 μW/cm^2^) for 48 h before being imaged using VA-TIRFM. The kymographs of the 120 s time course showed the residence of CLC2-RFP particles on the PM, indicated by white boxes (**A**). Quantitative analyses of CLC2-RFP particle lifetimes are shown (B: n = 241 and 39 images for each treatment). (**C**,**D**) UV-B-treated (40 μW/cm^2^, 24 h or 48 h pretreatment) and WL-grown 5-day-old *35s:GmNARK-GFP* (in *Arabidopsis*) specimens were incubated in 200 µM of BFA for 2 h. Internalization signal of GmNARK-GFP-termed BFA body was quantified as a ratio comparing each GmNARK-GFP signal on the PM (D: n = 142, 74, and 109 cells from left to right). The comparison ratio of the internalization signal under UV-B relative to WL is shown as percentage (**D**). (**E**,**F**) Four-day-old *35s:GmNARK-GFP Arabidopsis* seedlings were subjected to UV-B-treatment (10 μW/cm^−2^, 0, 2 d, 3 d, or 5 d pretreatment). Total GmNARK-GFP protein and AtRemorin1.1 levels and microsome protein were tested using Western blot. Proteins were analyzed via immunoblotting with anti-GFP, anti-Remorin1.1, and anti-RPN6 antibodies. RPN6 was used as a loading control. *uvr8-6* is *Atuvr8* mutant *Arabidopsis* (**E**). Quantification of GmNARK-GFP and REM1.1 protein levels was conducted. Protein signal ratios were used to indicate the signal strength of GmNARK-GFP and REM1.1 relative to each RPN6 signal on the protein blot, measured in arbitrary units via densitometry (**F**). Scale bars, 10 µm (**C**); error bar = S.D. *p*-values were determined using two-tailed Student’s t-test assuming equal variances; **** *p* < 0.0001).

## Data Availability

The raw data supporting the conclusions of this article will be made available by the authors on request.

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
