# Peer review of "UV-B Radiation Disrupts Membrane Lipid Organization and Suppresses Protein Mobility of GmNARK in *Arabidopsis"

_plants, 2024, doi:10.3390/plants13111536_

Round 1
Reviewer 1 Report
Comments and Suggestions for Authors
The paper by Liu et al presents an interesting effect of UVB light on plasma membrane organization and mobility of PM or sub-PM particles. The use advanced microscopy quantitative analysis to show differences in protein marker abundance, diffusion and membrane lipid order. Most of the data is solid with quantitation, but there are several issues that need to be addressed as indicated below.
-Statements regarding significance should be accompanied by quantitation and statistics. In line 103 and 104, only Fig 1A,B is quantified, but not Fig 3E.
-Methods are lacking details, specially on image analysis. For Fig 1, for example, there is no information regarding thresholding or segmentation. If not, the data in Fig 1d or would be overwhelmed by the dark pixels from the vacuole. How is “GP value” calculated? What is an “HSB picture”?
-Writing needs to be much improved, specially in the introduction and discussion sections.
-Lines 253-254 and 196-197: “UVB resulted in accumulation of a proportion of GmNARK proteins along the membrane”. Which proportion? High proportion or low proportion? Compared to? If this is referring to an increase in protein accumulation in the membrane, then there should be a quantification of the fluorescence intensity to demonstrate that.
-If a protein is being detected as a GFP fusion, the text should indicate it as such. The imaging is not reporting on the untagged native protein. For example, in instances describing the localization or abundance by fluorescence microscopy, “GmNARK-GFP” should be used instead of GmNARK.
-For VA-TIRFM; How are “bright” and “weak” particles defined?
-Fig 3C: BFA bodies should be co-labeled with an appropriate marker that labels endosomes (FM4-64. TGN marker or others) to identify them as such and differentiate them from endocytic or exocytic vesicles. Also, methods describing the quantification of these images is completely lacking. Were any ROIs selected and how?
All figures: define the n used for statistics (cells, images, plants, organelles?).
Other minor issues:
- Fig 1: Panel 1a “long term” is not describe in legend
- Define HSB
-This sentence is unclear (lines 119-120): The MBCD treatment “restored the promotional effect”…
-Care should be taken of listing the appropriate references throughout the text. For example, line 101 and 102 list ref 24, 25, which do not discuss how remorins are “required for assembly of nanodomains”.
Comments on the Quality of English Language
Editing of the introduction and discussion is needed.
Reviewer 2 Report
Comments and Suggestions for Authors
Article
The manuscript „UV-B radiation disrupts membrane lipid organization and suppress mobility of soybean nodule autoregulation receptor” by Liu et al. concerns the subject of how UV-B radiation influences the formation of ordered lipid domains on the plasma membrane and its effect on the Glycine max nodule autoregulation receptor kinase (GmNARK) protein. Studies were conducted on Arabidopsis thaliana plants with introduced 35s:GmNARK-GFP construct and exposed to UV-B radiaton.
The research idea is good, but in my opinion the manuscript needs to be corrected before publication.
Both the Title and the abstract should be corrected, because they indicate that studies were made on soybean plants while they were conducted on A. thaliana plants.
The Introduction part of the article need to be improved as it should include explanation why A. thaliana plants were used? The aim of the study is missing and it should be clearly indicated. The last part of the Introduction (line 82 – 92) is more of a summary and it could be written as part of conclusions.
In my opinion Results should be combined with Discussion part. In present form the results are poorly described and it is more like discussion than results description. So, results description should be extended to be more precise. The discussion should be improved as in my opinion it is somehow repetition of results. More references are needed. Further research directions should be indicated. For now it seems to be unfinished section.
The experiments were conducted with valid methodologies, but it should be explained why such methods were used? It is not clear. Only microscopic observations and calculations were included? Why? All performed analyses should be described in details in Materials and Method section.
In Materials and Methods section information about Methyl-β-cyclodextrin and Brefeldin A treatments is missing. There is also no information about the GP value and the Remorin1.2 signal calculations.
The protein accumulation analysis description is missing in Material and Methods. The protein extraction, electrophoresis, Western blot, immunoblot should be described in details in this section. The information included in Figure 3 capture is insufficient.
Conclusions should indicate some further directions of the research.
After implementing the proposed corrections, I may consider accepting the manuscript for publication. Other issues requiring improvement are as follows:
- Please once again carefully read the article because I found a lot of missing words (e.g. line 68 – ‘there still unknown’; line 73 – ‘may modified’, where ‘be’ is missing; line 74 ’which induced’; line 81 – ‘that is UV-B’ – ‘why’ is missing; line 91 – ‘UV-B stress (?) alters’; line 105’..level after 5 days UV-B exposure’, etc.) and typos (e.g. line 124 – ‘exposure’ or ‘exposed’; line 180 – ‘disrupted’ or’ disruption’?; line 186 – ‘efficient’ or ‘efficiency’?).
- The aim of the study should be clearly indicated at the end of the Introduction part.
- Line 125 – how the Remorin1.2 value was calculated?
- Line 130 – how the GP value was calculated?
- Line 131 – in the case of Figure 1f - why six values of n are presented if on the figure only four are presented?
- Line 152 and 160 – the authors refers to ‘Figure 2G’, but there is no such letter on figure 2.
- Line 264 – 266 – this sentence is difficult to understand, it should be corrected.
- Line 270, 279 – The Latin name of the plants should be written in italics.
- Line 271 – Celsius degree is missing when temperature is indicated.
- Paragraph 4.2 – more details about the method are needed, some references could be added.
- Line 302, 303, 305, 308, 311 – there is a sign (spiral circle) that is difficult to understand, did you mean µm?
- Paragraph 4.5 – Detailed information about the statistical test that was used should be added. Moreover the information about how results significance is marked should be included.
- Please check if the references are written according to journal recommendations.
Round 2
Reviewer 1 Report
Comments and Suggestions for Authors
Most of my comments were addressed by the authors, except the identity of BFA compartments.
Reviewer 2 Report
Comments and Suggestions for Authors
The authors followed my comments and made appropriate corrections. In my opinion, the manuscript can be accepted for publication in its current form.
Comments on the Quality of English LanguageIn my opinion, the quality of the English language is good. I found several typos and missing words, so the authors should carefully read the entire text again and make corrections.